

# Selection of quality indicators for nutritional therapy in pediatrics: a cross-sectional study conducted in Brazil

Julia Bertoldi[1,3], Aline Ferreira[2], Luiza Scancetti[2,3] and Patricia Padilha[2,3]

[1] Laboratório de Ciências do Exercício, Universidade Federal Fluminense, Niterói, Rio de Janeiro, Brazil
[2] Instituto de Nutrição Josué de Castro, Universidade Federal do Rio de Janeiro, Rio de Janeiro, Rio de Janeiro, Brazil
[3] Instituto de Puericultura e Pediatria Martagão Gesteira (IPPMG)/Programa de Residência Multiprofissional em Saúde da Criança e do Adolescente, Universidade Federal do Rio de Janeiro, Rio de Janeiro, Rio de Janeiro, Brazil

Corresponding authors
Julia Bertoldi,
julia_bertoldi@hotmail.com
Patricia Padilha,
patricia@nutricao.ufrj.br

## ABSTRACT

**Background:** Quality indicators for nutritional therapy (QINT) are important in assessing care and monitoring of resources. Among the 30 indicators proposed by International Life Sciences Institute, Brazil, there is still no evaluation of the most pertinent for Pediatrics.

**Objective:** To list the 10 main quality indicators for nutritional therapies (QINTs) for Pediatrics.

**Methods:** This was a two-phase cross-sectional study. Firstly, a questionnaire was answered by physicians, nutritionists, nurses, and pharmacists, all with having experience in nutritional therapy (NT) with Pediatrics, in Rio de Janeiro, Brazil. Participants assessed four attributes of QINT by using the Likert scale. A Top 10 ranked QINT list for Pediatrics was established. To verify the consistency of the questionnaire, Cronbach's Alpha coefficient was calculated. Secondly, the opinions of the participants on the results that were obtained were requested and the percentages of the positive responses were calculated.

**Results:** A total of 33 professionals participated in the first phase and 92% ($n = 23$ of 25) in the second phase approved the results of the selected indicators. Among the Top 10 QINTs, the three main ones were: #1: "Frequency of diarrhea in those patients on enteral nutrition" (mean = 13.194; $\alpha = 0.827$); #2: "Frequency of dietary nutritional prescriptions upon the hospital discharge of the NT patients" (mean = 12.871; $\alpha = 0.822$); #3: "Frequency of the NT patients who recovered their oral intake" (mean = 12.839; $\alpha = 0.859$).

**Conclusion:** When considering the consistency and the concordance that were obtained, it can be suggested that the list of Top 10 QINTs as proposed in this study will help in the evaluation of NT quality indicators for Pediatrics.

## CLINICAL RELEVANCY STATEMENT

The constant self-evaluation for the quality of nutritional therapy (NT) that is used in each health service is necessary, as a way to monitor this important part of health care and to optimize the material and human resources that are used. A Top 10 list of quality indicators for nutritional therapy (QINT) in Pediatrics was configured as being a standardization of the most pertinent quality indicators for nutritional therapies (QINTs) within a pediatric context, in order to facilitate the monitoring of NT for Pediatrics in daily clinical practices.

## INTRODUCTION

Health quality indicators are used as a way of measuring and evaluating the quality of care to be provided and with a purpose of optimizing the human and material resources employed, directing attention to the critical stages that are in need of greater control (*Bittar, 2001*). By using indicators for health care quality monitoring enables one to demonstrate the cost-benefit results that justify the material investment and the employment of human resources. This is especially relevant for those practices that are essential for the success of the treatments, with important inherent costs, such as NT (*Waitzberg, 2010*).

In order to monitor NT services, as well as to redefine the goals for correcting possible processes and to achieve quality, the Clinical Nutritional Task Force of the International Life Sciences Institute, Brazil (ILSI Brazil) published in 2008, a book entitled "Quality Indicators for Nutritional Therapy," which included a list of QINTs, as proposed by health professionals from all over the country that were directly involved in clinical practices using NT (*Waitzberg, 2008*). In 2010, a new publication brought a revised list of QINTs that contained the Top 10 QINTs that were considered to be the most useful, simple, objective, and low cost, as listed by NT practitioners and health professionals. The current list has a total of 30 indicators, which is further divided into eight categories, according to the focus that they give to the analysis of NT quality provided. These listed categories are: Category A: general aspects; Category B: nutritional assessment; Category C: NT indications; Category D: preparations—pharmaceutical evaluations, handling, quality control, conservation, and transportation; Category E: administration—access roads; Category F: administration—calories and proteins; Category G: clinical and laboratory control; and Category H: final evaluations (*Waitzberg, 2010*).

Within the context of NT for Pediatrics, there are a lack of specific QINT recommendations and NT practices are often evaluated by several parameters without any uniformed criteria. *Valete et al. (2009)* used the following parameters in order to analyze the quality of NT that was received by 203 underweight preterm infants from a public maternity hospital in the city of Rio de Janeiro, Brazil: (a) days to enter an enteral diet; (b) days to reach a full enteral diet; (c) days to start and the time of parenteral nutritional use; and (d) the type of diet upon hospital discharge.

*Tume, Latten & Darbyshire (2010)* evaluated enteral nutritional therapy (ENT) practices with critically ill children from the time of starting ENT, the infused caloric supply when compared to the estimated nutritional needs, as well as the interrupted time

of the enteral diet. *Botrán et al. (2011)* related the severity of clinical conditions for critically ill children with biochemical parameters and with measurements of energy expenditure prior to the beginning of ENT. *Ricciuto, Baird & Sant'anna (2015),* when comparing the prolonged use of a nasogastric catheter in relation to gastrostomy, as well as when evaluating the ENT practices in a pediatric hospital, collected the data of: anthropometrical measurements, the rate of aspiration pneumonia, intercurrences in oral feeding and problems related to probe obstructions.

It should be noted that studies on NT for Pediatrics evaluate the quality of NT in different ways. They do not take into account the existence of QINT as has been standardized by the ILSI, since they have chosen to focus on adult care (*Waitzberg, 2010*). Therefore, it has now been proved to be relevant to analyze and to propose the most appropriate QINTs for Pediatrics, thus guiding the various services in their choice. Accordingly, the aim of the present study has been to list the Top 10 QINTs for Pediatrics through a questionnaire that was applied to medical professionals, nutritionists, nurses, and pharmacists, all having an experience of NT in Pediatrics.

## MATERIALS AND METHODS

A total of 37 professionals from the categories that make up the multidisciplinary nutritional therapy team (MNTT) in Brazil (*Brasil, 2000*) were invited to participate in the study (*Brasil, 2000*). Twelve nutritionists, 10 nurses, nine pharmacists, and six physicians, who were linked to public and/or private hospitals in the city of Rio de Janeiro, Brazil, participated. The eligibility criteria demanded that they worked and/or specialized in NT for Pediatrics.

After explaining the outline and the purpose of the study, those professionals who accepted to participate were asked to sign the Informed Consent Term. This study was carried out with the approval of the Institutional Research Ethics Committee (Certificate of Presentation for Ethical Assessment—CAAE: 48629615.1.0000.5264).

This was a cross-sectional study that was divided into two phases. The first stage consisted of sending or delivering a questionnaire that was composed of objective and closed questions. The professionals were asked to punctuate each one of the 30 QINTs, divided into eight categories as proposed by ILSI in 2010 (*Waitzberg, 2010*), with the attributes of "useful," "simple," "objective," and "low cost." These attributes were mentioned by *Bittar (2001)* as being necessary for any health quality indicators. The scores were based on the Likert scale, a psychometric scale by which the professional opinion can be quantified. According to his or her own opinion, his or her agreement or disagreement for each QINT was attributed by assigning grades from 0 to 4, as follows: 0, when he/she totally disagreed; 1, when he/she disagreed; 2, when he/she did not agree or disagree (indifferent); 3, when he/she agreed; and 4, when he/she totally agreed (*Boone & Boone, 2012*).

The second phase of the research was carried out in order to inform the participating professionals with the questionnaire's results and to obtain their opinions. To this end, a list of the Top 10 QINTs for Pediatrics was delivered by e-mail, followed by two questions, for which the answers should be "Yes" or "No:" "Do the Top 10 QINTs for Pediatrics reflect your own opinion?" and "Are you satisfied with the Top 10 QINTs that were chosen

for the pediatric population?" From the answers, the percentages of the professionals that responded positively to the results were calculated.

## Statistical analyzes

The definition of a minimum sample size was based on the methodology as proposed by *Verotti et al. (2012)*, which aimed to evaluate the best QINTs for adult individuals. In this particular case, a total of 24 participants were recommended as being a sample size capable of generating statistically significant results.

For analyzes of the data, the information that was collected by the questionnaire that was used in the first stage was entered into Excel® software. The scores for each QINT were summed and the average sum of the opinions from all of the professionals was calculated for each individual QINT.

In order to verify the consistency and the reliability of the results that were obtained from the opinions of the different professionals, Cronbach's Alpha coefficient was calculated. The consistency of the questionnaire was considered to be excellent when the calculated Cronbach's Alpha value was >0.9; it was good when ≤0.9 and >0.8; it was acceptable when ≤0.8 and >0.7; it was questionable when ≤0.7 and >0.6; it was poor when ≤0.6 and >0.5; and it was unacceptable when ≤0.5 (*Gliem & Gliem, 2003*). The mean variance for each QINT was also calculated.

The 10 most scored QINTs were identified from the 10 largest arithmetical averages of the four attributes, depending on the reliability values. Data analyzes were performed by using the statistical package for social sciences for Windows Version 23.0. The results are presented as averages, with Cronbach's Alpha values and Variance.

# RESULTS

## Characterization of the participants

A total of 37 professionals with experience and/or with a specialization of NT in Pediatrics were invited to participate in the study, of which 12 subjects were nutritionists, 10 subjects were nurses, nine subjects were pharmacists, and six subjects were physicians. Of the 12 invited nutritionists, all accepted and answered the questionnaire. Of the invited nurses, nine of the 10 accepted and answered the questionnaire. Of the nine invited pharmacists, one refused the invitation and two accepted, but these two did not respond to the questionnaire, making a total of six. In relation to the physicians, all six that were invited accepted to participate and they all answered the questionnaire. Altogether, 33 professionals contributed to the evaluation of QINTs for Pediatrics, 94% of them were from public hospitals and 6% were from private hospitals.

## Evaluation of QINTs for Pediatrics

The results of the first phase are shown in Table 1. The table presents the QINTs in descending order of the obtained mean scores, the respective QINT category cited by *Waitzberg (2010)*, the results of Cronbach's Alpha values and the variances.

Regarding the second phase that consisted of 33 participating professionals, 25 gave their opinion about the results of the 10 best evaluated indicators. A 92% approval rating

**Table 1  Classification of QINTs in Pediatrics evaluated by the multidisciplinary professionals.**

| Rank | QINT | Category (*Waitzberg, 2010*) | Mean | Cronbach's α | Variance |
|---|---|---|---|---|---|
| 1 | Frequency of diarrhea in patients on ENT | G | 13.194 | 0.827 | 0.089 |
| 2 | Frequency of nutrition dietary prescription for patients on NT at discharge | H | 12.871 | 0.822 | 0.076 |
| 3 | Frequency of patients on NT who recovered oral intake | A | 12.839 | 0.859 | 0.037 |
| 4 | Frequency of admission BMI measurement in patients on NT | B | 12.839 | 0.711 | 0.030 |
| 5 | Frequency of periodic reevaluation of nutritional planning in NT | B | 12.742 | 0.814 | 0.076 |
| 6 | Frequency of patients with altered glycemia on ENT and PNT | G | 12.741 | 0.681 | 0.140 |
| 7 | Frequency of nutrition probe obstruction in patients on ENT | E | 12.226 | 0.816 | 0.017 |
| 8 | Frequency of abdominal distension episodes in patients on ENT | G | 12.161 | 0.873 | 0.062 |
| 9 | Frequency of nutritional screening in hospitalized patients | B | 12.129 | 0.869 | 0.133 |
| 10 | Frequency of constipation episodes in patients on ENT | G | 12.065 | 0.940 | 0.037 |
| 11 | Frequency of fasted patients before the onset of NT | A | 12.032 | 0.870 | 0.058 |
| 12 | Frequency of conformity of ENT indication | C | 12.032 | 0.747 | 0.258 |
| 13 | Frequency of nonconformities related to time of preparation, transport and storage in ENT | D | 11.903 | 0.722 | 0.274 |
| 14 | Frequency of patients with peripheral NPT lasting more than seven days | D | 11.774 | 0.795 | 0.004 |
| 15 | Frequency of patients with high gastric residue on ENT | G | 11.613 | 0.906 | 0.018 |
| 16 | Frequency of patients with central PNT lasting less than seven days duration | D | 11.581 | 0.837 | 0.012 |
| 17 | Frequency of digestive fasting for more than 24 h in patients on ENT or oral nutrition | B | 11.452 | 0.913 | 0.007 |
| 18 | Frequency of CVC infection in patients on PNT | E | 11.452 | 0.753 | 0.178 |
| 19 | Frequency of inadvertent exit of enteral catheter in patients on ENT | E | 11.419 | 0.839 | 0.035 |
| 20 | Frequency of patients with hydroelectrolytic changes on PNT | G | 11.355 | 0.796 | 0.210 |
| 21 | Frequency of phlebitis due to insertion of peripheral venous catheter in patients on PNT | E | 11.323 | 0.856 | 0.039 |
| 22 | Frequency of measurement or estimation of energy expenditure and protein needs in patients on NT | B | 11.226 | 0.786 | 0.353 |
| 23 | Frequency of days of adequate energy administration in patients on total NT | F | 10.871 | 0.835 | 0.200 |
| 24 | Frequency of medical care for patients on NT | A | 10.677 | 0.614 | 0.061 |
| 25 | Frequency of hepatic dysfunction among patients on ENT and PNT | G | 10.323 | 0.846 | 0.384 |
| 26 | Frequency of renal dysfunction in patients on ENT and PNT | G | 10.129 | 0.871 | 0.369 |
| 27 | Frequency of biochemical assays in nutritional assessment in patients on ENT | B | 9.935 | 0.816 | 0.538 |
| 28 | Frequency of SGA application in patients on NT | B | 9.710 | 0.915 | 0.105 |
| 29 | Frequency of nutritional recall in patients on NT | B | 9.581 | 0.895 | 0.194 |
| 30 | Frequency of pneumothorax due to CVC insertion for PNT | E | 9.194 | 0.841 | 0.012 |

Notes:
BMI, body mass index; CVC, central venous catheter; ENT, enteral nutritional therapy; NT, nutritional therapy; PNT, parenteral nutritional therapy; QINT, quality indicator for nutritional therapy; SGA, subjective global assessment.
Categories according to *Waitzberg (2010)*: A: general aspects; B: nutritional assessment; C: NT indications; D: preparations—pharmaceutical evaluations, handling, quality control, conservation and transportation; E: administration—access roads; F: administration—calories and proteins; G: clinical and laboratory control; and H: final evaluations.

was found (23 participants stated that the Top 10 chosen QINTs reflected their own opinion and they were satisfied). Of these participants, 10 were nutritionists, five were nurses, six were pharmacists, and four were physicians.

## DISCUSSION

The list of Top 10 QINTs for Pediatrics that is presented here brings together perspectives from four different categories of professionals: nutritionists, nurses, pharmacists, and physicians, and high values of Cronbach's Alpha Coefficient were found. This study has indicated that there was a good consistency among the responses to the questionnaire and that there was a good agreement of opinions, since the Top 10 Alpha values were predominantly ranged from 0.9 to 0.8. The high percentage of participants who approved the list (92%) reinforced this view, especially when we analyzed the absolute numbers. This was since only two of the 25 participants who responded to the report reacted negatively to the results.

In the present study, the 10 best QINTs that were evaluated belonged to five of the eight indicator categories, with the predominance of those referring to clinical and laboratory control (Category G) and nutritional assessments (Category B), with four and three indicators in the Top 10, respectively. These categories were the ones that had the most QINTs, with eight indicators each. The other categories that had representatives in the Top 10 were general aspects (Category A), administration: access roads (Category E), and final evaluations (Category H). This showed a satisfactory diversity of categories in the Top 10, probably because it was built by four different professional classes.

In a study by *Verotti et al. (2012)*, the Top 10 elected QINTs belonged to four different categories, being highlighted by Category E—administration: access roads, with three of the five indicators of this category being in the Top 10. In our study, only one indicator of this category was among the 10 best scored, occupying the seventh position: "Frequency of nutritional probe obstructions in patients using ENT," which was also present in the Top 10 for adults (*Verotti et al., 2012*). Probe obstructions can occur due to a variety of causes, such as diets with increased viscosity and the type of probe material, with polyurethane the most indicated. In addition, this can also happen by the formation of precipitates from drug–nutrients and drug–drug interactions or by an improper administration of macerated drugs, for example (*Lord, 2003*). These situations can and should be controlled and corrected in order to maintain the qualities of NT, since diet interruptions are reduced for clogged probes exchange.

"Frequency of patients with altered glycemia when using ENT and parenteral nutritional therapy (PNT)" was also well evaluated, ranking sixth in both studies (*Verotti et al., 2012*). Changes in glycemia may be multifactorial, occurring in situations with diseases such as the critical conditions of pancreatitis, diabetes mellitus, liver cirrhosis, as well as in errors when estimating the patient's energy needs or the slow progression of diet volumes (*Van Den Berghe et al., 2003*). Whatever the causes, these should be monitored and corrected for better patient health care.

"Frequency of diarrhea in patients when using ENT" was also well evaluated in both studies, ranking second in the Top 10 for adults and first in the Top 10 for Pediatrics (*Verotti et al., 2012*). Diarrhea, that is characterized by the presence of three or more liquid feces per day, may reflect a microbiological inadequacy in the diet, a high osmolarity and/or a high infusion rate. However, it may also be due to infections of another nature or

by a prolonged use of medications, such as antibiotics. Whatever the causes, these should be monitored and corrected for better patient health care. In a study by *Afonso (2016)* that was carried out in a reference center for oncology treatments in Rio de Janeiro, Brazil, they found an association between the presence of diarrhea and an increased hospitalization time for the children and the adolescents in an Intensive Care Unit for the treatments of malignant neoplasia when receiving ENT. In the same study, it was also verified that an abdominal distension increased the time of mechanical ventilation, indicating that this variable was also important. This indicator of "Frequency of abdominal distension episodes in patients when using ENT" was present in our study in eighth place.

The indicator named "Frequency of conducting nutritional screening in hospitalized patients" was also common in the two Top 10 lists for adults and children. The list by *Verotti et al. (2012)* presented the mentioned indicator in first place and, in the present study, it was in ninth place. Nutritional screening, which is understood to be the initial identification of those patients suffering from a nutritional risk, has a great contribution for the quality assurance of NT, as it guides the priority actions for adequacy and the supply of nutritional needs. In general, nutritional screening is quickly implemented through questionnaires that can easily be applied by a health team. Subjective global assessment (SGA) is one of the possible types of nutritional screening, because it is simple, easy, inexpensive, and non-invasive. However, it is a method that is applicable only for the adult and elderly populations (*Detsky et al., 1987*; *Fontoura et al., 2006*). In the present study, the indicator entitled "the frequency of an SGA application in NT patients" was poorly evaluated, ranking 28th, while it ranked 10th in the QINT list for adult (*Verotti et al., 2012*).

In 2007, *Secker & Jeejeebhoy (2007)* developed the Pediatric subjective global nutritional assessment, which used the data of anthropometry, dietary intake, physical examinations, functional capacities, gastrointestinal symptoms, and the degree of metabolic stress. *Saraiva et al. (2016)* performed a transcultural translation and an adaptation of this particular questionnaire, with a subsequent validation. It is the only validated nutritional screening instrument for the Brazilian pediatric population to date. The nutritional screening method that best suits each health service should be chosen jointly by the MNTT. Today, the Brazilian Ministry of Health recommends the use of STRONGkids (screening tool for risk on nutritional status and growth), although it does not have a validation in Brazil yet. STRONGkids consists of four steps: subjective clinical assessments, the presence of high-risk disease or major surgery, ingestion, and losses in recent days, as well as a referred loss or an insufficient weight gain (*Brasil, 2016*; *Hulst et al., 2010*).

"Frequency of measurements of the body mass index (BMI) at admission in patients on NT" was well evaluated, ranking fourth in the Top 10 pediatric list. However, it should be emphasized that in many clinical situations, it is not easy to measure a BMI calculation accurately, as in the cases of muscular spasticity of encephalopathic patients, or those that are restricted to bed. In addition, studies with children suffering from cancer have observed that the evaluation of an isolated BMI may underestimate malnutrition when
compared to other forms of body composition assessments (*Barr et al., 2011*; *Lemos, De Oliveira & Caran, 2014*). In these and other situations, the team should analyze, be flexible and choose the QINTs that would be more appropriate for a pediatric population regarding these particular cases.

The limitations of this study should be considered. Although there were more participants than the minimum necessary to generate statistically significant results, there were a limited number of medical and pharmaceutical professionals participating. However, the statistical calculations corrected these differences, and the high Cronbach's Alpha values obtained indicate that there is a good consistency and concordance among the specialists opinions, despite of their different number and work places. In addition, it should be noted that these professionals work in hospitals in the city of Rio de Janeiro, not reflecting the reality of other cities and regions of Brazil and/or of that of the world. In view of the above, it can be suggested that pediatric health services use the list of Top 10 QINTs, that have been evaluated here by our multidisciplinary team, as a reference. They should choose with a critical appreciation of those indicators that are more pertinent for their service, while combining simplicity, usefulness, objectivity, and low cost.

From the necessity of evaluating the NT quality provided in pediatric contexts, recently in 2017, the Pediatric Nutrition task force of ILSI Brazil has announced a proposal of 13 QITNs in pediatrics (*Gandolfo et al., 2017*). Some of these proposed indicators are different from those previously published by ILSI (*Waitzberg, 2008*, *2010*) and, consequently, different from those on the present study, which are the most 10 relevant among the variety of 30 indicators, according to the opinion of multidisciplinary group of professionals. Although the publication of 2017 (*Gandolfo et al., 2017*) does not exhibit the methodological process for the indicators selection, they are also possible options for analysis and use by pediatric services as convenient and applicable to the local care service characteristic.

For the Top 10 QINTs in pediatric care, new studies are proposed in order to evaluate such NT qualities by the use of these QINTs. In addition, this should also occur by associating the frequencies of these phenomena that have been evaluated by the QINTs for the incidences of morbimortality and by making possible, propositions for the cut-off points for QINTs in Pediatrics, when related to positive outcomes.

## CONCLUSION

Owing to the good consistency and the agreements that were found in the data analyzes, it can be stated that the Top 10 QINTs for Pediatrics represent the opinions of a multidisciplinary team of specialists and/or of experienced nutritionists. The fact that 10 of the 30 indicators as proposed by the ILSI were chosen directly by them, makes more practical routine evaluations. In this way, it could guide health services to evaluate the NT qualities in a more homogeneous way. However, it should be noticed that this study have been developed with professionals that work in Rio de Janeiro, which could result a point of view of an specific health care reality. Thus, further studies considering other regions are required. Nevertheless, it is suggested that pediatric health services consider the best indicators that have been found by the present study for their decisions on how to assess the NT quality performed.

## TRANSPARENCY DECLARATION

The lead author affirms that this manuscript is an honest, accurate, and transparent account of the study being reported. The reporting of this work is compliant with STROBE guidelines. The lead author affirms that no important aspects of the study have been omitted and that any discrepancies from the study as planned have been explained.

## ABBREVIATIONS

| | |
|---|---|
| **QINT** | Quality Indicators for Nutritional Therapy |
| **QINTs** | Quality Indicators for Nutritional Therapies |
| **NT** | Nutritional Therapy |
| **ENT** | Enteral Nutritional Therapy |
| **ILSI** | International Life Sciences Institute, Brazil |
| **MNTT** | Multidisciplinary Nutritional Therapy Team |
| **SGA** | Subjective Global Assessment. |

## ACKNOWLEDGEMENTS

The authors thank all the professionals for contributing to the construction of the Top 10 QITNs for Pediatrics list. This work counted with no sources of funding. The authors declare no potential conflicts of interest.

### Funding

The authors received no funding for this work.

### Competing Interests

The authors declare that they have no competing interests.

### Author Contributions

- Julia Bertoldi conceived and designed the experiments, performed the experiments, analyzed the data, contributed reagents/materials/analysis tools, prepared figures and/or tables, authored or reviewed drafts of the paper, approved the final draft.
- Aline Ferreira performed the experiments, analyzed the data, contributed reagents/materials/analysis tools, authored or reviewed drafts of the paper, approved the final draft.
- Luiza Scancetti performed the experiments, analyzed the data, authored or reviewed drafts of the paper, approved the final draft.
- Patricia Padilha conceived and designed the experiments, performed the experiments, analyzed the data, contributed reagents/materials/analysis tools, authored or reviewed drafts of the paper, approved the final draft.
## Human Ethics

The following information was supplied relating to ethical approvals (i.e., approving body and any reference numbers):

This study was carried out with the approval of the Institutional Research Ethics Committee (Certificate of Presentation for Ethical Assessment—CAAE: 48629615.1.0000.5264).

## Data Availability

The raw data has been supplied as a Supplemental File.

## Supplemental Information

Supplemental information for this article can be found online at http://dx.doi.org/10.7717/peerj.4630#supplemental-information.

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
