# Peer review of "Selection of quality indicators for nutritional therapy in pediatrics: a cross-sectional study conducted in Brazil"

_PeerJ, doi:10.7717/peerj.4630_

## Round 0.1 · original submission · Major Revisions

Please carefully consider all the suggestions raised by the reviewers.

·

Basic reporting

No comment

Experimental design

As for the rigorous performance of the methodological course I have some observations.
I understand as a fragility of the study the participants' choice for vocational training, different places of work (private and public system) and not considering the disparate number among them.

Just as a suggestion, I believe that if it were a comparative cross-sectional study between different populations, the results would be less questionable.

However, the choice of multiplicity of actors and sites does not invalidate the research.

"When you put different professions and different places of action you can say that we can be faced with multiple views and different angles", that is, I believe that in the continuity of this study we must stratify the qualification of the indicators according to place of performance and type of professional activity.

Validity of the findings

It`s OK
Only, the conclusion can be improved by identifying the need for further study..

Additional comments

Congratulations!
I believe that this study could be further developed so that it becomes a reference for other services to point out their best quality indicators in Pediatric Nutritional Therapy.
The experience in this area brings us multiple glances and we certainly need these to better serve those who so badly need this type of therapy performed effectively, efficiently and quality.

Reviewer 2 ·

Basic reporting

English: Recommend the english be revised by a professional native english scientific editor
Literature - There is a publication on the topic of quality control in clinical nutrition in pediatrics, from the ILSI force task on pediatric nutrition, from March 2017, available on line at http://ilsibrasil.org/publication/indicadores-de-qualidade-em-terapia-nutricional-pediatrica/ which should be quoted and discussed by the authors
Article structure - Pertinent to the area
Results and hypotheses - no comment

Experimental design

Aims and scope - no comments
Knowledge gap - At ILSI - Brasil , it is available a document written by the pediatric clinical nutrition task force, under the chairman of Dr Rubens Feferbaum, that suggest the pediatric IQTN
rigor and ethics - no comments
Methods - no comments

Validity of the findings

Replication - The authors may want this study to be replicated by all the pediatric nutrition support groups in Brazil, in order to validate their preliminary findings
Data - Suffers from the bias of being done only in Rio de Janeiro state, and by invitation, what is a selection bias of participants
Conclusions: The authors may want to rephrase the conclusions that would be applicable for Rio de Janeiro state, and invite validation

Additional comments

The authors should be commented by trying to stablish the top 10 quality control indicators for pediatric clinical nutrition
Some modifications are important to be included , in order to improve the understanding and scope of the manuscript

Reviewer 3 ·

Basic reporting

No comment

Experimental design

No comment

Validity of the findings

No comment

Additional comments

283/5000
The following correction is suggested for the present study: In Table 1, if categories of Qualitative Indicators in Nutritional Therapy refer to the same ones mentioned in the introduction and published by Waitzberg DL, for better understanding they should be described in methods.

---

## Round 0.2 · accepted · Accept

I think the manuscript has improved with the modifications included by the authors, so I consider it is now acceptable for publication.

Reviewer 2 ·

Basic reporting

ok

Experimental design

ok

Validity of the findings

The authors have answered adequately all my questions, no further comments

Additional comments

The authors have answered adequately all my questions, no further comments